# Sensory discrimination of chemical and temperature stimuli in the acoel *Symsagittifera roscoffensis*

Nikita Komarov[1], Christopher Aeschbacher[1], Laurent Sauterel[1], Evan Zuercher[1], Xavier Bailly[2]*, Pedro Martinez[3,4]*, Simon G. Sprecher[1]*

1 Department of Biology, University of Fribourg, Fribourg, Switzerland, 2 Multicellular Marine Model (M3) Lab, Station Biologique de Roscoff, CNRS—Sorbonne University—Place Georges Teissier, Roscoff, France, 3 Department de Genètica, Microbiologia i Estadística, Universitat de Barcelona, Barcelona, Spain, 4 ICREA (Institut Català de Recerca i Estudis Avancats), Barcelona, Spain

* simon.sprecher@unifr.ch (SGS); pedro.martinez@ub.edu (PM); xavier.bailly@sb-roscoff.fr (XB)

## Abstract

Environmental cues provide critical sensory information for the survival of animals. Understanding how distinct sensory cues elicit or modulate certain behaviour thus provides insights into the adaptations to rapid and continuous changes in the surrounding world. Intertidal ecosystems are particularly exposed to environmental fluctuations. Due to changing exposure to seawater, animals are subjected to continuous fluctuations of temperature or salinity during the course of day-night and tidal cycles. Animals in intertidal environments show physiological and behavioural adaptations to these changes. Acoel worms constitute an important component of intertidal ecosystems. *Symsagittifera roscoffensis*, a well-studied species inhabiting the Atlantic coastline, has been extensively described in terms of its anatomy and development, yet its physiological responses remain poorly understood. When the acoel *S. roscoffensis* are exposed to daylight during the tidal cycle, these animals are found at the surface of sandy beaches, which enhance the exposure of their photosynthetic algal symbionts to light. Moreover, *S. roscoffensis* shows a strong positive phototaxis as well as both positive and negative geotaxis, both being evolved behavioural adaptations to enhance light exposure for its photosymbiont. Currently little is known about other sensory systems and their functions in this, or any other acoel worm. In this study, we probe sensory capabilities of *S. roscoffensis* focusing on chemical and temperature cues. Using two-choice and barrier assays, our findings support that *S. roscoffensis* shows avoidance behaviours to increased temperature and salinity, preferring cooler environments with lower salinity. We demonstrate that early branching bilaterians possess the sensory capacity to identify specific chemical and environmental stimuli, adding to the knowledge that may prove useful in understanding marine ecosystems in a period of global climate change that greatly affects aquatic environments.

**Data availability statement:** All relevant data are within the paper.

**Funding:** The work of the current manuscript has been supported by the Swiss National Science Foundation (grant 310030_219348 and IZKSZ3_218514) to SGS. The funders had no role in study design, data collection and analysis, decision to publish, or preparation of the manuscript.

**Competing interests:** The authors have declared that no competing interests exist.

## Introduction

While many animals utilise external sensory organs to determine the nature of their surroundings, the compositional, morphological and anatomical organization of sensory organs varies greatly in different animal clades, including different, often unrelated genes being responsible for the same functions, indicating convergent evolution [1]. Sensory systems evolved to transduce signals of their surroundings into neuronally encoded information, which are subsequently integrated and processed in the central nervous system to ultimately elicit or modulate the response of an animal.

The highly variable environmental physical and chemical conditions of intertidal ecosystems, which are submerged at high tide and exposed to air at low tide, likely provide crucial sensory cues for animals. While desiccation as well as fluctuations in temperature and salinity are stressors for animals living in the intertidal zones, certain physiological adaptations also provide advantages to optimise exposure to light. One example of such specialisation can be found in several intertidal inhabitants such as the acoel worms. The phylogenetic position of acoels, along with the fact that some species live in close association with algae, has made them of particular interest in the fields of evolutionary developmental biology (evo-devo) and comparative physiology. However, most existing studies on these animals focus on developmental and anatomical aspects, with relatively few addressing their physiology or the influence of environmental conditions on behavior, survival, or the dynamics of the host–algae association. To address this gap, we initiated a series of experiments aimed at testing the effects of various chemicals on these animals. Most of the compounds we tested—such as variations in salinity and temperature—are highly relevant to the rapidly changing conditions in coastal environments, where many acoel species naturally occur. In this context, understanding the response of intertidal acoels should illuminate previously unknown aspects of their physiology. We have selected, as mentioned, the acoel flatworm *Symsagittifera roscoffensis* who lives in the Atlantic Coast and is currently subjected to different environmental, human-derived, challenges. This millimetre-scale worm lives in the intertidal zones of the Atlantic coasts of Europe, in a photosymbiotic relationship with the algae species *Tetraselmis convolutae* [2], which it locates at the juvenile stage through unknown means, and has captivated marine research since the 19th century [3,4]. Phylogenetically, it belongs to the phylum Xenacoelomorpha [5] that diversified very early on within the Bilaterian lineage [6], though the precise phylogenetic placement of Xenacoelomorpha remains debated. As one of the few well-known bilaterian organisms capable of photosynthesis through photosymbiosis, *S. roscoffensis* challenges some conventional metabolic principles such as reliance on the symbiont for amino acid (AA) synthesis [7], born from the need to integrate processes occurring in two organisms under changing (daily or seasonally) environments, all of which has resulted in significant attention garnered from the fields of marine biology, ecology, and physiology [8,9]. Although the algal symbiont supplies *S. roscoffensis* with most of its energy and nutrients [10], it is still unclear whether the acoel also needs to obtain extra food from its surroundings (*i.e.*, whether it is partly mixotrophic). If so, it is not yet known what chemical

cues are relevant or how a chemosensory system might guide it to those external resources. Some classical studies have examined the influence of environmental conditions on the survival of Symsagittifera roscoffensis and, in this context, on the establishment and maintenance of its symbiosis with algal partners [11–15],

In many animals, chemosensation is mediated by specialised external organs such as insect antennae, vertebrate olfactory epithelia, the antennules and aesthetascs of crustaceans, or the amphid sensilla of nematodes [1]. These organs bear high densities of receptor neurons exposed to the medium and thus are considered "canonical" chemosensory structures. By contrast, no such obvious external chemosensory organs have been described in *S. roscoffensis,* except a some suggested through ultrastructure analysis, as well as some putative receptor genes [16–18].

Like other acoels, the *S. roscoffensis* possesses paired rhabdomeric eyes: cup-shaped invaginations lined with micro-villar photoreceptors that detect light direction and intensity; and a statocyst, an internal, fluid-filled vesicle containing a statolith whose deflection under gravity provides the *S.roscoffensis* with information on body orientation [19–22].

While the strong phototactic behaviour as well as geotaxis of *S. roscoffensis* has been well investigated [21,23], comparatively little is known about other sensory capacities. Bridging this gap in knowledge may uncover how non-visual sensory modalities govern how *S. roscoffensis* locates and retains its algal symbiont and selects microhabitats in a highly variable intertidal environment. Establishing these capacities also provides comparative data for the early evolution of multimodal sensory integration in bilaterians.

Here, we try to remedy this lack of information by investigating the chemosensory and thermosensory sensitivities of *S. roscoffensis* by examining a variety of chemical and environmental modalities to determine the functional significance of identified sensory sensilla [16,24].

Given that this worm lives in an aquatic, intertidal, environment, where changes of physical parameters or chemical composition are most probable (e.g., light intensity, UV radiation, temperature, oxygen, and salinity [25]), we hypothesised that *S. roscoffensis* should be capable of sensing not only environmental stimuli such as temperature, and salinity, but are also able to respond to chemical cues, including potentially those emitted by their internal symbionts (and their free-living relatives) or other species occupying their same ecological niche.

## Results

The chemosensory behaviours in *S. roscoffensis* remains currently understudied. In order to gain insights into the chemosensory capacity of *S. roscoffensis* we first performed two-choice assays to assess navigational responses when exposed to putatively relevant chemical cues. As first set of experiments we investigated the chemosensory behavioural response to a set of AA known to be used as neurotransmitter-precursors or neurotransmitters in both invertebrates and vertebrates [26]. While the dependence of *S. roscoffensis* on its symbiont to provide essential neurotransmitters remains unknown, it has been previously shown that glutamate and aspartate, two AAs that may act as neurotransmitters [27], were only detected in *S. roscoffensis* with symbionts, while it was absent in juvenile worms that did not yet harbour its 1 algae [28]. Other AAs that act as neurotransmitters such as glycine, or neurotransmitter precursor AAs such as tyrosine (dopamine and noradrenaline precursors), tryptophan (serotonin precursor), were not tested in regard to navigational response in previous studies. Given the conserved nature of these neurotransmitter systems in metazoans [26], it can be presumed that they are also present in Acoels. Hereby, we tested whether *S. roscoffensis* is able to detect and respond to the presence of these AAs (glutamate, aspartate, tryptophan, tyrosine and glycine) in the environment (water column), as there is no evidence that tryptophan, tyrosine, and glycine are provided by the symbiont, using a two-choice navigation assay, measuring preference indices at 2-, 5-, and 10-minute timepoints. We placed *S.roscoffensis* on dishes containing agarose (control), or dishes containing half agarose and half agarose supplemented with the given AA (Fig 1B). During 10-minute trials, we observed that the preference indices at the 10-minute timepoint for glutamine (Mean PI = −0.001, SEM = 0.103, Brown-Forsythe ANOVA: 4.909df1, df2 = 4, p = 0.9996) and tyrosine (Mean PI = −0.1, Brown-Forsythe ANOVA: 4.909df1, df2 = 4 p = 0.74) largely match those of the plain agarose control, suggesting *S. roscoffensis* did not

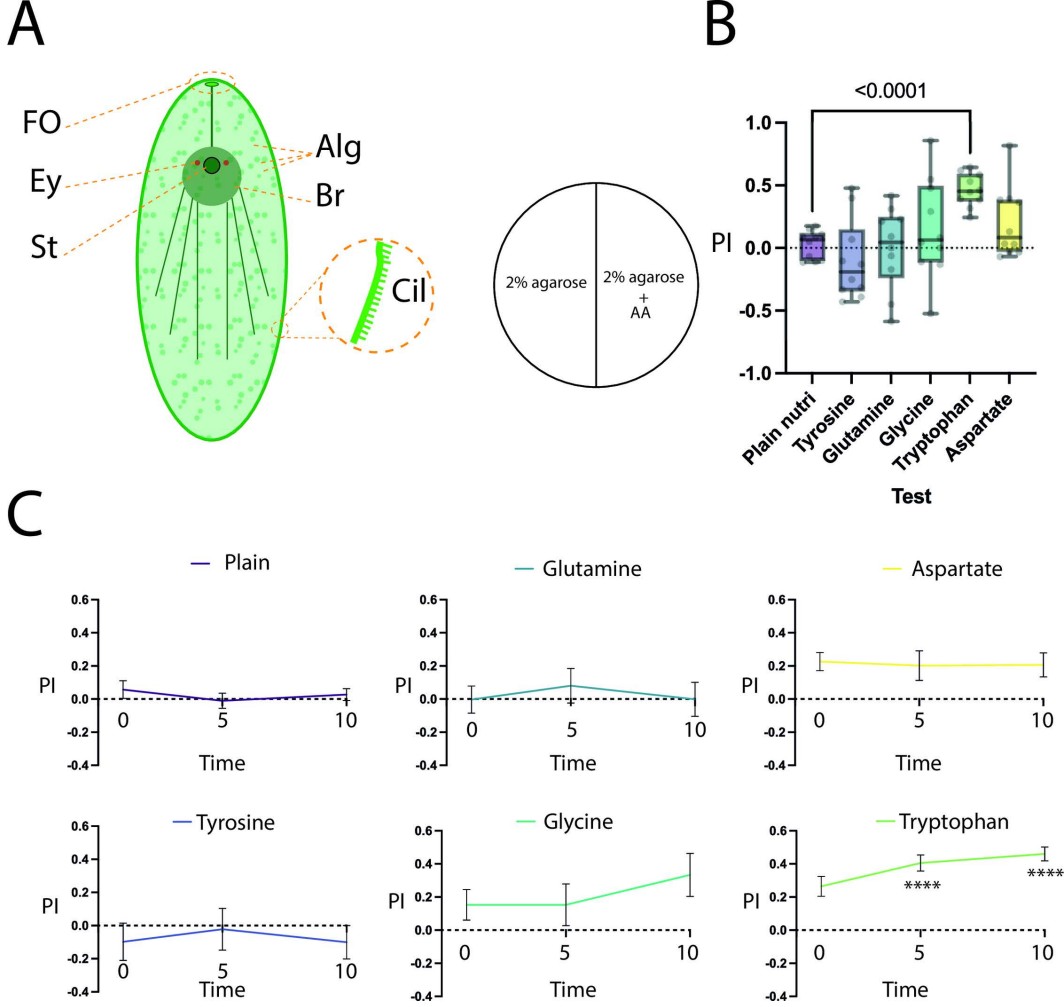

**Fig 1. Chemosensory preference of *S. roscoffensis* to neurotransmitter-related amino acids.** A: Overview of *Symsagittifera roscoffensis* outlining the symbiont algae (Alg), brain (Br), eye spots (Ey), the gravisensory statocyst (St) and the putatively chemosensory frontal organ (FO). B: left: diagram of the behaviour arena indicating a two-choice set-up with one half containing plain agarose, and the other half containing agarose with dissolved test substance, in this case the indicated AA. Right: preference indices for indicated test AAs compared with the "nutrient plain" agarose control (purple) at the 5-minute timepoint, all trials displayed with min-max bars. C: Summary of the PIs shown as time-series across tested AAs, mean with SEM. N = 10 for all tests. **** - p < 0.0001, ns where not shown, Brown-Forsythe and Welch ANOVA test with Dunnet's T3 multiple comparisons test against the control group.

prefer either side (PI = 0.03). For the AA aspartate, we observed a mild preference, albeit not showing statistically significant differences from the control (Mean PI = 0.2, SEM = 0.0895, Brown-Forsythe ANOVA: 4.909df1, df2 = 4 p = 0.36, Fig 1B, 1C). For glycine, we observed that there was a moderate preference across longer timepoints (PI = 0.15, SEM = 0.126, Brown-Forsythe ANOVA: 4.909df1, df2 = 4 p = 0.86, Fig 1C). Tryptophan, however, evoked the strongest response, with significance being reached above the 5-minute timepoint (PI = 0.46, SEM = 0.0417, Brown-Forsythe ANOVA: 4.909df1, df2 = 4, p < 0.0001 Fig 1B, 1C).

Considering the dynamic nature with varying salinity in intertidal waters due to evaporation and temperature changes during low tides, it has been shown that during low tides the salinity of sea water can change by up to 300% [29]. Since this is directly related to the environment of the acoel, we hypothesised that they may seek specific salinity levels,

suggesting an ability to sense this modality. Therefore, we asked whether changes in salinity of the substrate would result in a preference phenotype.

Here, *S. roscoffensis* were given a choice between a plain agarose/dH$_2$O substrate and agarose with dissolved artificial sea water (ASW) adjusted to 34 ppm salinity using a refractometer, to reflect the salinity of the ambient sea water measured outside the Roscoff marine station (Fig 2A). We observed one-fold ASW (34 ppm) showed a mildly positive, albeit insignificant, preference index at the 10-minute timepoint (Mean PI = −0.22, SEM = 0.092, Brown-Forsythe ANOVA: 7.268df1, df2 = 2, p = 0.09) compared with the environmental sea water control, and across all time points (Fig 2B). However, we observed that the PI for two-fold ASW concentration (68 ppm) was significantly negative across the time series, reaching significance after the 5-minute timepoint (Mean PI = −0.45, SEM = 0.0387, Brown-Forsythe ANOVA: 7.268df1, df2 = 2, p < 0.0001) (Fig 2B). The concentration-dependent responses to variations in salinity thus allow us to assume a sensory ability to perceive high salt concentrations. This potentially reflects an ability to adapt to the dynamic variations in the intertidal environments where *S. roscoffensis* is found.

The finding that *S. roscoffensis* appears to prefer lower salt concentrations suggests that this condition is preferential for the acoel. However, we wanted to evaluate whether *S. roscoffensis* decided to avoid higher salt concentrations, or

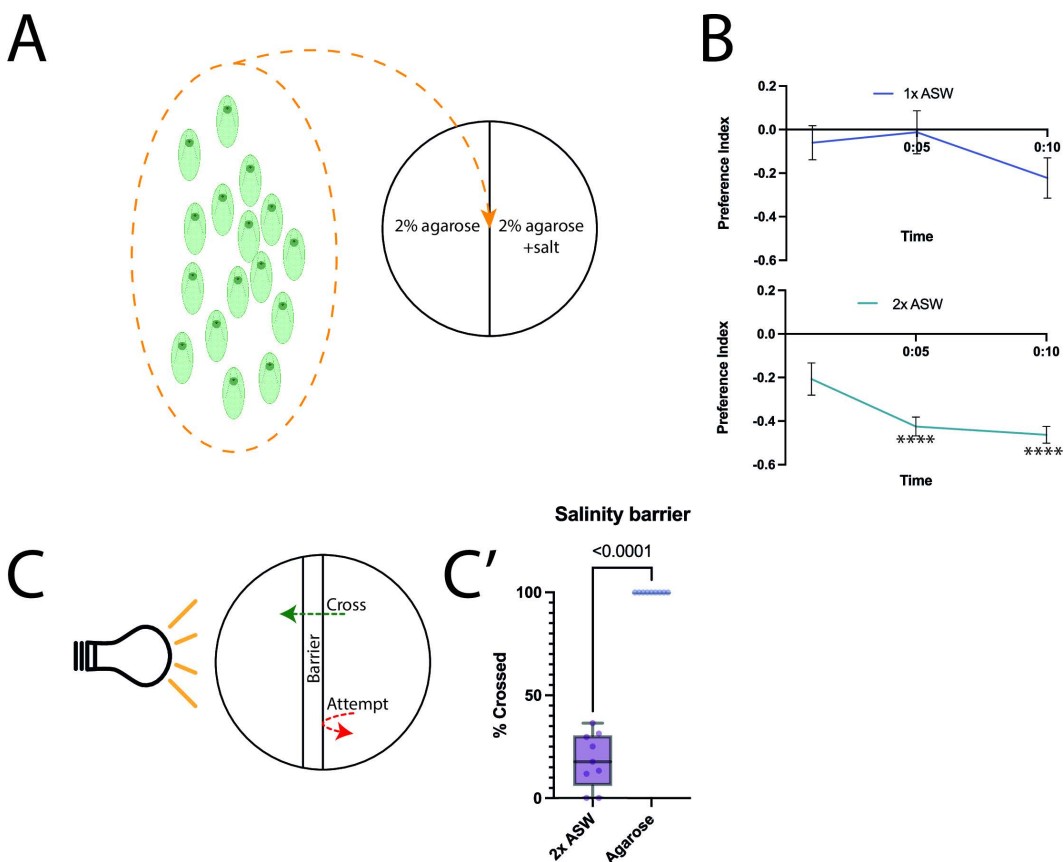

**Fig 2. Behavioural responses of *S. roscoffensis* towards varied salinity conditions.** A: overview of experimental conditions showing pooled *S. roscoffensis* being placed on the behaviour arena comprised of plain agarose on one half, and agarose with added salt content on the other half. B: Time-series diagrams showing preference indices over time, with twofold ASW concentration showing significance beyond the 5-minute timepoint; N = 10 for all trials, mean and SEM shown. C: salinity barrier assay diagram indicating the considered "attempt" and "cross" determinants; C': – quantification of successful crosses across the high-salinity barrier compared with a plain agarose barrier; N = 10 trials, all points and min-max bars are shown. **** - p < 0.0001, ns where not shown, Brown-Forsythe and Welch ANOVA test with Dunnet's T3 multiple comparisons tested against the control group.

whether another physiological effect, such as reduced mobility, was at play. To test this, we developed a barrier-assay, in which we took advantage of the known strong positive phototaxis behaviour of *S. roscoffensis.* In short, we removed a 1 cm agarose strip in the middle of the plain agarose plate and replaced it with agarose containing twofold ASW. Subsequently a group of *S. roscoffensis* was placed on one side of the strip while a LED lamp was placed, laterally, at the opposing side with the aim of inducing the acoels to cross the midline attracted by the light. We recorded their movement across the high-salinity barrier during phototaxis. Here, we observed that when presented with a plain (agarose in dH$_2$O) barrier, 73% (403/553) of *S. roscoffensis* attempted (evaluated by counting the encounter of *S. roscoffensis* with the barrier) to cross the barrier towards the light source, and 100% (403/403) of the attempts were successful, evaluated by the complete crossing of the barrier towards the light source. Meanwhile, when presented with a two-fold ASW barrier, 52% (184/352) of *S. roscoffensis* attempted to cross the barrier, with only 19.6% (36/184) of attempts resulting in a cross (Fig 2C), and significantly higher compared with the agarose barrier (N = 10. Agarose mean = 100 SEM = 0, 2xASW mean = 18.33 SEM = 4.417, Welch's t test t = 18.49, df = 8, p < 0.0001). Therefore, we show that despite the presence of a highly attractive light stimulus, *S. roscoffensis* avoid the approach it if they encounter high salinity, suggesting a hierarchical aversion-attraction perception and decision making.

Next, since it has been shown that juvenile *S. roscoffensis* are attracted by the symbiont and its supernatant [14,30], we asked if adult *S. roscoffensis*are similarly responsive to chemical cues emitted by symbiotic and non-symbiotic algae (which also live in their surrounding waters). Thus, to probe whether the presence or absence of the symbiont *T. convolutae* or other, closely related algal species (*C. concordia, T. chuii,* and *T. striata*) may be used as a sensory cue, we performed similar two-choice assays, providing *S. roscoffensis* a choice between plain agarose and either a fresh algal culture F/2 medium, or a supernatant of an extract of algal cultures, over 10-minute trials with preference indices being measured at the 2-, 5-, and 10-minute timepoints. We observed that while there is a strong preference for pure F/2 medium, mature *S. roscoffensis* strongly avoid the supernatants from cultures of *T. convolutae, C. concordia, T. chuii.* However, they show a slight but statistically insignificant negative PI to *T. striata* extracts (Fig 3A). Furthermore, we utilised the same barrier decision-making assay to determine whether the presence of the *T. convolutae* supernatant displays similar aversive properties as compared to excessive salinity. Here, we found that only 37% (252/687) of *S. roscoffensis* attempted to cross the barrier, with a total success rate of 19.4% (49/252) (Fig 3B, 3D) which is significantly lower than the agarose control (N = 10. Agarose mean = 100 SEM = 0, *T. convolutae* mean = 15.95 SEM = 5.810, Welch's t test t = 14.47, df = 8, p < 0.0001).

Apart from chemical environmental cues, temperature may act as a key determinant for an animal to navigate towards favourable conditions. Therefore, we chose to characterise whether *S. roscoffensis* displays also preference for specific temperature conditions. Here, we placed groups of *S. roscoffensis* on plates with different temperature gradients: a) high (30°C) to low (5°C), b) high to ambient (18°C), c) low to ambient, or d) ambient to ambient ranges (Fig 4A). We observed that there is, as expected, no difference in the side preference index between ambient-to-ambient areas at 10 minutes (PI = −0.01), and, similarly, no significant preference index between the low to ambient conditions compared to the ambient/ambient control at 10 minutes (Mean PI = −0.05, SEM = 0.08, Brown-Forsythe ANOVA: 5.592df1, df2 = 2, p = 0.9366). On the other hand, when presented with a choice between a high and low temperature, *S. roscoffensis* tend to show a positive PI to the low temperature side after 10 minutes compared with the ambient/ambient control (PI = −0.37, SEM = 0.074, Brown-Forsythe ANOVA: 5.592df1, df2 = 2, p = 0.0028). Moreover, *S. roscoffensis* preferred the side with ambient temperature in comparison to the side with higher temperature at the 10-minute timepoint compared with the ambient/ambient control (MeanPI = −0.30, SEM = 0.092, Brown-Forsythe ANOVA: 5.592df1, df2 = 2, p = 0.0398), suggesting an avoidance behaviour towards higher temperatures (Fig 4B, 4C). These experiments support the notion that *S. roscoffensis* do prefer lower temperatures over ambient or high temperatures. This in turn suggests that while low temperatures are neither attractive nor aversive, high temperature is decidedly an aversive cue, since animals show no preference between ambient and low temperatures, and only negative preference to high temperature.

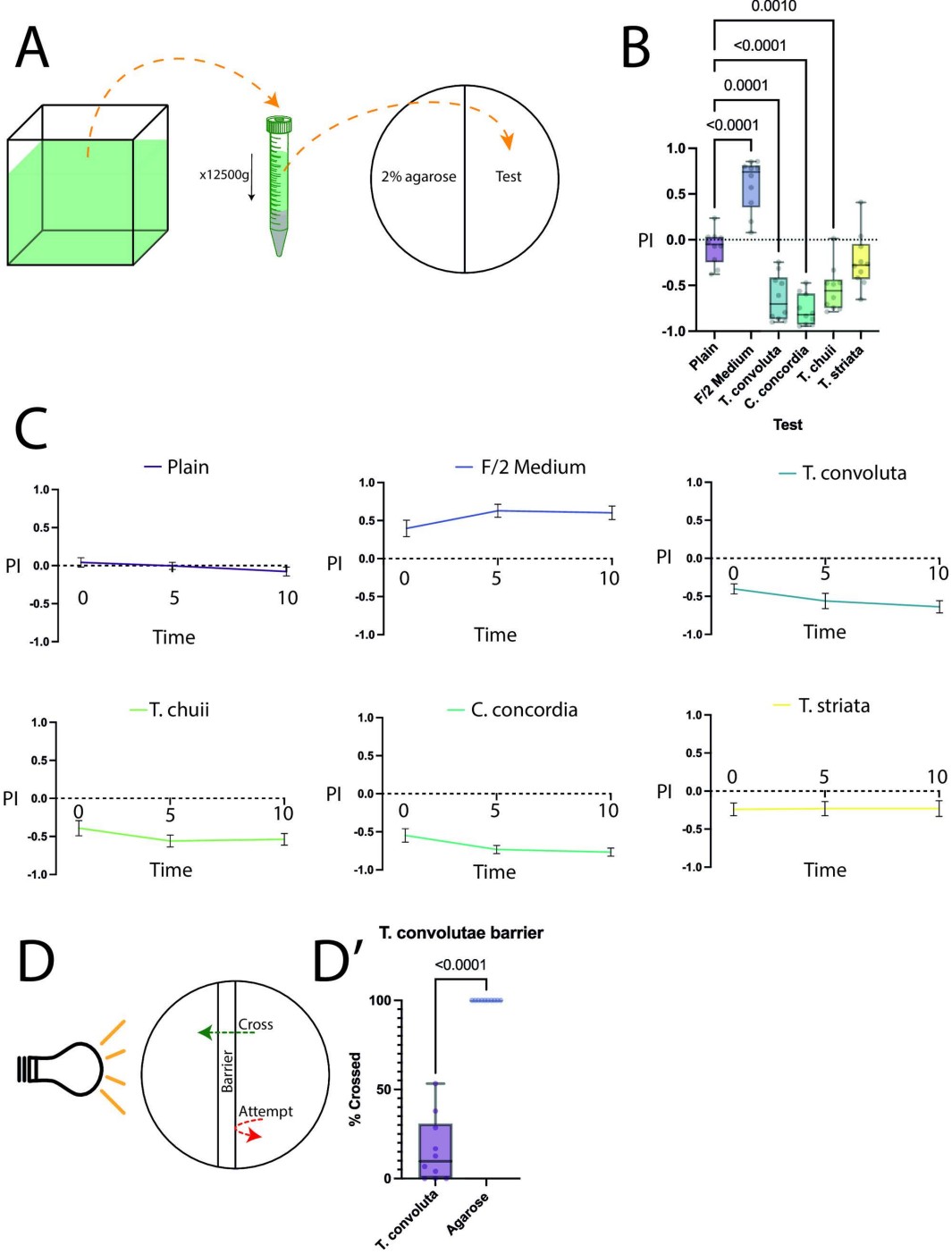

**Fig 3. Behavioural tests for algal symbiont extract.** A: diagram showing the process of extraction of algal culture. Approximately 20 ml of algal culture was Dounce homogenised, then spun down and supernatant was used to prepare the choice assay, with half of the plate containing plain agarose, and the other half containing agarose dissolved in extracted supernatant. B: summary of behaviour phenotypes at the 10-minute timepoint, with F/2 medium by itself showing a strong attractive index, while supernatant extracted from algal cultures shows strong aversive indices; N = 10 for all trials, all points and min-max bars shown. C: time-series diagrams showing preference indices over time. F/2 medium preference is positive beyond the 5-minute time-point, while supernatant from *T. convoliutae, T. chuii,* and *C. concordia* causes immediate strong aversion. While supernatant from *T. striata* also shows a mild aversion, it is not statistically significantly different from the plain control. N = 10 for all trials, mean with SEM shown. D: *T. convolutae* culture supernatant barrier cross assay performed similarly to the salinity barrier assay. D': *S. roscoffensis* was less likely to cross the barrier towards the light when compared with a plain agarose barrier.

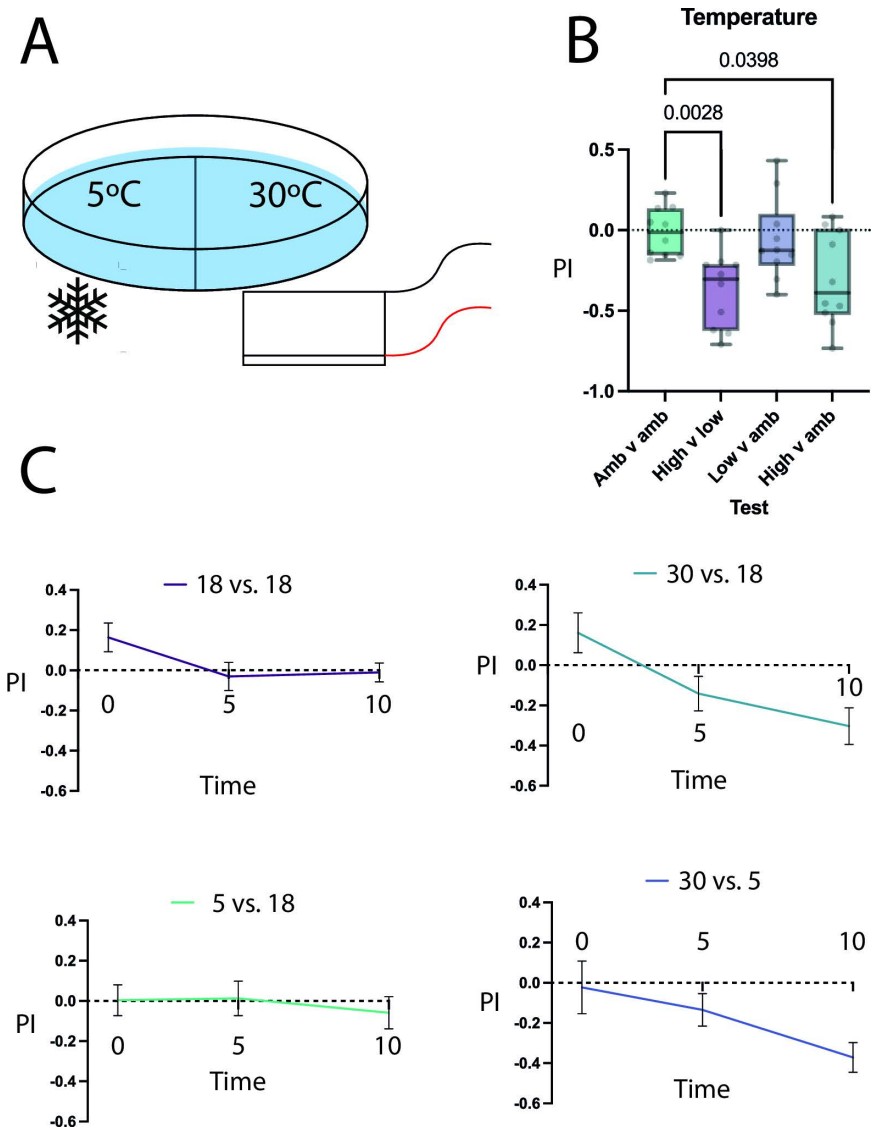

**Fig 4. Behaviour tests for varying temperature conditions.** A: diagram showing the experimental setup, with a plate being in contact with ice on one side, and a heated Peltier plate on the other. B: summary of behavioural phenotypes at the 10-minute timepoint, showing a general preference towards lower temperatures. C: individual test time-series for each condition, showing a trend for increased preference for cooler temperatures over time. N = 10 for all tests. Brown-Forsythe and Welch ANOVA test with Dunnet's T3 multiple comparisons test against the ambient control group.

## Materials and methods

### Animal collection and maintenance

Adult *S. roscoffensis* was collected on 01.09.2023 from wild colonies from the sands in the vicinity of Roscoff (48°43'38.6"N 3°59'16.5"W) and Carantec (48°40'14.4"N 3°54'40.3"W) (Brittany, France). The collected sand was then placed in deep trays and *S. roscoffensis* was collected with Pasteur pipettes during low tides when *S. roscoffensis* display negative geotaxis. Collected *S. roscoffensis* were transferred into plastic containers (approximately 1000–5000 animals per container) with filtered natural sea water, continuously flowing to ensure consistent oxygenation, at constant water temperature (14 ± 1°C) and constant air temperature (18°C)at Station Biologique de Roscoff (Roscoff, Brittany, France). All

experiments were performed during similar tidal and circadian conditions, between 09:00 and 15:00. All experiments were performed using animals with visible symbiont presence. Animals were not discriminated by sex. Experiments were performed between 03.09.2023 and 08.03.2023. Animals were not provided supplementary food outside of what may have been present in the surrounding water and sand.

**Two-choice behaviour assays**

All behaviour assays were performed using the method described in Maier et al., 2021, substituting *S. roscoffensis* for *Drosophila* larvae. In brief, 87 mm petri dishes were filled with 2% agarose diluted in distilled water and allowed to solidify for a minimum of 10 minutes. Following this, one half of the agarose layer was cut and discarded. Then, the now-empty half of the dish was filled with 2% agarose containing the substance of choice diluted in distilled water (for details, see below). *S. roscoffensis* were pooled in groups between 50–200 individuals and placed in the centre of the plate, with enough sea water being added to form a complete, thin, layer on top of the agarose. Experiments were monitored and the number of *S. roscoffensis* in each half-plate were photographed and counted after 2, 5 and 10 minutes of each trial (total 10 trials for each experiment shown). The different timepoints were selected to see an immediate response (2 minutes) and up to 10 minutes to ensure sufficient exploration time of the arena by the animals. The temperature was kept constant at ambient (18°C) ensuring no heating or cooling sources, as well as direct sunlight, were present. The plates were rotated 180° between each individual N to ensure that any potential directional lighting effect was mitigated. The position of the experimenter during each trial remained constant, with the rotation of the plates serving to mitigate any effect this may have caused. *S. roscoffensis* on each side were then manually counted in a blinded fashion (*i.e.,* the counter was not aware of which experiment, timepoint, or the stimulus side they were counting). No monitoring for activity was performed during the trials, and a preference index was calculated using the following formula:

$$PI = \frac{N \text{ animals on test half} - N \text{ animals on plain half}}{N_{total} \text{ animals}}$$

   The chemicals used were of the highest purity available, as follows:

   Amino Acid assays: L-Glutamine (Sigma-Aldrich, 56-85-9), L-Aspartic acid (Sigma-Aldrich, 56-84-8), L-Tyrosine (Sigma-Aldrich, 60-17-4), L-Tryptophan (Sigma-Aldrich, 73-22-3), L-Glycine (Roth, 3187.3). All tests were performed at 20mM concentration.

   Artificial sea water (Aquarium Sytems, Instant Ocean) diluted in distilled water to a concentration of 34 ppm (onefold) or 68 ppm (twofold).

   Choice assays with culture supernatant from the classical and facultative symbiont: *Tetraselmis convolutae* (the classical symbiotic algae of *S. roscoffensis*), and *Tetraselmis striata*, *Tetraselmis chuii* and *Chlamydomonas concordia*, (three facultative symbionts of *S. roscoffensis*) were cultured in flasks containing of sea water and F/2 feeding medium (UTEX culture collection of algae) for a period of 15 days. Before collecting the supernatant, the algae were examined at the microscope to exclude any abnormalities or increased cell death. The content of the flasks was collected by means of a Douce tissue grinder following 20 passes of the pestle, before centrifuging the slurry. 87 mm petri dishes were prepared with one half containing 2% agarose in distilled water and the other half containing 2% agarose in a 25 ml mix composed of 12 ml distilled water and 13 ml supernatant solution of one of each four algae species, respectively.

**Barrier assays with twofold salinity and culture supernatant from the classical symbiont**

87mm dishes were prepared with 2% agarose in distilled water on both sides, with a 10 mm strip cut out from the middle, and subsequently filled with agarose containing twofold ASW or *T. convolutae* supernatant. An LED lamp (OSRAM LED, 80012 White) was placed at a 45° angle, 9 cm above the table at a distance such that the lamp-oriented side of the dish

was illuminated at 1100 lux, the middle of the plate 840 lux and the side opposite of the lamp with 400 lux. *S. roscoffensis* were placed on the side of the barrier further from the light, and subsequently observed over the course of 10 minutes, with the attempts and successful crosses counted. 10 minutes were given for each trial in order to negate any effects of reduced mobility, which was not noticed by the experimenter.

### Thermal preference assays

87mm petri dishes were coated with 2% agarose before being placed in a Styrofoam box for thermal insulation. One half of the petri dish was in contact either with ice water (reaching a temperature of 5°C), RT (18-21°C) or with a Peltier plate running at 24W, reaching a temperature of 30°C. The temperatures were selected to ensure sufficient difference between conditions under practical limitations. were pooled and placed in the middle of the petri dish in a similar fashion to the choice assays described above. The number of *S. roscoffensis* was then recorded on each side, and a preference index was calculated.

### Statistical analysis

All statistical analysis was performed using GraphPad Prism 10. For each trial of each experiment, a fresh set of acoels was used, thus no worms were subjected to multiple stimulation experiments. For time trials, each time point was treated independently. Welch's test was used to determine unequal variance within the individual experiments, and, combined with a relatively low sample size (n = 10), Dunnet's T3 multiple comparison test was chosen. Individual statistical results are presented in the relevant results sections and figure legends.

### Permits and site access

No permits were required for this work on invertebrate model animals. Site access to the Roscoff Marine Biology Station was provided and supervised by Xavier Bailly (xavier.bailly@sb-roscoff.fr).

### Data availability

All behaviour data are available on Zenodo: https://zenodo.org/records/15470219?token=eyJhb-GciOiJIUzUxMiJ9.eyJpZCI6ImRiYTc3OTM0LWNkMDItNDVkMC04NzY4LWE0N2FiNDQ0NjNhO-SIsImRhdGEiOnt9LCJyYW5kb20iOiJiYWQ5ZmU2NGI3ODdhMzM5YzhhY2I1ODUzODE1YmFhZSJ9.KJhEgc7RN0yJZeKa9npMKW3wBQx8lxdL_XoUNSqaPo8gY6xI2Djula4wxGVZrwfyDPuLm9oTamO9RGZXOahk4w.

## Discussion

Intertidal animals are exposed to variable environmental conditions, whether daily or along the seasons [25]. These conditions have generated specific adaptations, reflected in their physiologies and their behavioural responses. Here we investigated sensory capabilities in relation to chemical and physical cues of the acoel flatworm *Symsagittifera roscoffensis*, an acoel living in obligatory symbiosis most of its life. Under laboratory conditions, *i.e.,* agarose substrate vs natural sand, our findings indicate that the acoel *Symsagittifera roscoffensis* can display distinct behavioural preferences when exposed to a range of chemical and environmental stimuli. Furthermore, the variability of chemicals used in our assays allows the uncovering of varied sets of phenomenologies. The 20mM concentration for AAs was chosen as it is the standard for small animal behaviour, such as in the *Drosophila* larva [31,32], or even in vertebrates, where this concentration is taken as the average that is discriminated readily by the animals [33]. It has been shown to provide sufficient responses without being toxic. While higher than the physiological concentrations or ones found in regular food, the diffusion through agarose in general is slower than in water. We do not believe that diffusion rates will differ between different AAs, as the average size is around 4Å, while agarose gel pores typically range between 30-200nM. Other factors, such as ion

concentrations may be more relevant, thus for our experiments concentrations were kept equal. For instance, the differing responses to different AAs raise a question about the mechanistic nature of this response. One explanation of the positive preference towards Tryptophan, a precursor to serotonin and melatonin, may be due to either a direct behavioural decision to seek it out due to regular scarcity, or alternatively, it may be due to a continuous need in the bioavailability of serotonin in the brain, where it could modulate (as in other invertebrates) basic aspects of their physiology such as feeding, olfactory sensitivity, egg-laying, and homeostasis.

To isolate responses exclusively to the stimuli of interest, our experimental conditions therefore differ from the natural environment in which *S. roscoffensis* is typically found. For instance, the substrate used in our experiment was agarose, as opposed to the natural substrate of sand, which may contain a large variety of chemical and biological contaminants that could, in turn, affect the behaviour of *S. roscoffensis*. However, for the purposes of this study, we aimed to observe any response specifically to a narrow range of individual, ecologically relevant stimuli, rather than observing *S. roscoffensis*' behaviour in the natural environment. Furthermore, we maintained a constant ambient temperature, which may not be the case in nature due to factors such as winds, waves, or clouds. Further studies may be performed to understand the impact of these factors on the responses of *S. roscoffensis* to varied stimuli. For example, future work could use sand as a substrate, perform chemosensory experiments under varying temperature and light conditions, or test at different points of the tidal cycle.

The aversion to increased salinity may also relate to the naturally varying changes in the habitat of the worms. During low tide, as water evaporates from the surface of the sand, salinity will increase, thus causing a changing environment for *S. roscoffensis*. Thus, it may be presumed that *S. roscoffensis* would prefer a specific level of salinity to remain in an isotonic environment and avoid a hypotonic one. However, it is known that *S. roscoffensis* emerges to the surface during low tide conditions, thus contraindicating an aversion to increasing salinity levels. One explanation for this may be that the strong negative geotaxis, but not positive phototaxis responses, overpower the natural aversion to increased salt levels, and the worms instead fine-tune their positioning within different levels of salinity on the surface.

The results for the responses to the symbiont supernatant also provide a surprising new perspective on the interactions between these host (worm) and the algae. While it is known that juvenile *S. roscoffensis* incorporate algae into their bodies, suggesting a level of attraction to any chemical cues that the algae may emit, adults show the opposite response. Firstly, we controlled for whether it is the medium of the algal culture that causes a behavioural phenotype, and, indeed, we found that it is an appetitive stimulus. However, despite this, the presence of any algal emissions, and not the algae themselves, causes a strongly aversive response. It is possible to speculate that once *S. roscoffensis* reach a state of maturity, including incorporating enough algae within them (with a narrow stoichiometry), a physiological switch occurs which causes them to avoid the free-living alga in their immediate environment. Perhaps, this may be in order to avoid overcrowding and allow new or growing juveniles more access to the algae, thus ensuring that the population is maintained, and self-competition is avoided. On the other hand, considering that in these experiments algal extract was used, the aversive behaviour may be due to *S. roscoffensis* avoiding broken (thus dead) algae, perhaps being expelled by dead *S. roscoffensis*. The presence of certain chemicals may thus indicate that the locality is under stress, as indicated by the liberated chemicals and metabolites from broken algae, indicating that *S. roscoffensis* need to hide or migrate to safer places (e.g., DMSP, dimethylsulfoniopropionate (an organosulfur compound) [34]). Further studies on adult and juvenile *S. roscoffensis* may provide the answers to these questions. For example, the concentration present in our tests may have been too high, resulting in the aversive response we observe, thus performing a concentration dilution series, using additional or distantly related algae, or controlling for algal cell number may prove as an interesting study in the future.

Like the responses of the salinity assays, the phenotype of high-temperature avoidance may also play a role in the navigation of *S. roscoffensis* on the surface during low tide. Since increasing temperature causes increased evaporation of the water, this may be an interlinked response with the avoidance of high salinity. Furthermore, higher temperatures may result in increased metabolism within the algae, or the worms themselves. Increased metabolism may result in increased

stress, for example from excess oxygen production resulting in elevated oxidative stress. Thus, avoiding increased temperatures may be a physiological response to avoid excess or premature death. Contrarily, lower temperatures may instead result in a neutral preference index due to either a lack of ability to detect such temperatures, or the fact that the temperature of the water fluctuates between an average low of 9°C to an average high of 17°C across the year. Thus, the "low temperature" condition in our assay may not be sufficiently lower than the temperature experienced by *S. roscoffensis* in the natural environment. While in [23]an increased stress in both very low (0° C) and very high temperatures (30° C) compared to the normal condition was shown, the worms were still able to survive. It is therefore reasonable to postulate that the worms tend to seek an environment within the optimal temperature range of their physiology. In this context, it is important to point out that our work represents a good complement to the one published by Thomas and collaborators [23], where they test the response of *S. roscoffensis* to several physical and chemical cues. Though they use the same species, their stimuli are different than those studied by us (in their set up: light, vibration, sea water or frozen worms), bringing the study of behaviour in acoels under a more secure footing. In all, we show that early-branching bilaterians display sensory responses to a variety of stimuli, and further studies can be performed to understand how the natural environment can affect the sensory responses of *S. roscoffensis*.

## Author contributions

**Conceptualization:** Nikita Komarov, Christopher Aeschbacher, Laurent Sauterel, Evan Zuercher, Xavier Bailly, Pedro Martinez, Simon G. Sprecher.

**Formal analysis:** Nikita Komarov, Christopher Aeschbacher, Laurent Sauterel, Evan Zuercher.

**Funding acquisition:** Simon G. Sprecher.

**Writing – original draft:** Nikita Komarov, Simon G. Sprecher.

**Writing – review & editing:** Nikita Komarov, Xavier Bailly, Pedro Martinez, Simon G. Sprecher.

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
