## [Decision Letter · Decision Letter 0]

15 May 2025

Dear Dr. Sprecher,

Thank you for submitting your manuscript to PLOS ONE. After careful consideration, we feel that it has merit but does not fully meet PLOS ONE’s publication criteria as it currently stands. Therefore, we invite you to submit a revised version of the manuscript that addresses the points raised during the review process.

We look forward to receiving your revised manuscript.

Kind regards,

Phuping Sucharitakul

Academic Editor

PLOS ONE

**Journal Requirements:**

1. When submitting your revision, we need you to address these additional requirements. Please ensure that your manuscript meets PLOS ONE's style requirements, including those for file naming. The PLOS ONE style templates can be found at https://journals.plos.org/plosone/s/file?id=wjVg/PLOSOne_formatting_sample_main_body.pdf and https://journals.plos.org/plosone/s/file?id=ba62/PLOSOne_formatting_sample_title_authors_affiliations.pdf 2. In your Methods section, please provide additional information regarding the permits you obtained for the work. Please ensure you have included the full name of the authority that approved the field site access and, if no permits were required, a brief statement explaining why. 3. Thank you for stating the following financial disclosure: grants:-310030_219348 -IZKSZ3_218514to SGS by the Swiss National Science Foundation  Please state what role the funders took in the study.  If the funders had no role, please state: "The funders had no role in study design, data collection and analysis, decision to publish, or preparation of the manuscript." If this statement is not correct you must amend it as needed. Please include this amended Role of Funder statement in your cover letter; we will change the online submission form on your behalf. 4. Thank you for stating the following in the Acknowledgments Section of your manuscript: We would like to express our gratitude to the Swiss National Science Foundation (SNSF) for providing the funding for this project (grant 310030_219348 and IZKSZ3_218514 to SGS). We are also thankful to the University of Fribourg (UniFr), members of the Department of Biology and the Marine Station in Roscoff for their support and help throughout our research. We also thank all members of the Sprecher Lab for their invaluable feedback on our experimental design, procedures, and reporting.We note that you have provided funding information that is not currently declared in your Funding Statement. However, funding information should not appear in the Acknowledgments section or other areas of your manuscript. We will only publish funding information present in the Funding Statement section of the online submission form. Please remove any funding-related text from the manuscript and let us know how you would like to update your Funding Statement. Currently, your Funding Statement reads as follows: grants:-310030_219348 -IZKSZ3_218514to SGS by the Swiss National Science Foundation  Please include your amended statements within your cover letter; we will change the online submission form on your behalf. 5. When completing the data availability statement of the submission form, you indicated that you will make your data available on acceptance. We strongly recommend all authors decide on a data sharing plan before acceptance, as the process can be lengthy and hold up publication timelines. Please note that, though access restrictions are acceptable now, your entire data will need to be made freely accessible if your manuscript is accepted for publication. This policy applies to all data except where public deposition would breach compliance with the protocol approved by your research ethics board. If you are unable to adhere to our open data policy, please kindly revise your statement to explain your reasoning and we will seek the editor's input on an exemption. Please be assured that, once you have provided your new statement, the assessment of your exemption will not hold up the peer review process. 6. Please review your reference list to ensure that it is complete and correct. If you have cited papers that have been retracted, please include the rationale for doing so in the manuscript text, or remove these references and replace them with relevant current references. Any changes to the reference list should be mentioned in the rebuttal letter that accompanies your revised manuscript. If you need to cite a retracted article, indicate the article’s retracted status in the References list and also include a citation and full reference for the retraction notice.

Reviewers' comments:

**Comments to the Author**

1. Is the manuscript technically sound, and do the data support the conclusions?

Reviewer #1: Yes

2. Has the statistical analysis been performed appropriately and rigorously?

Reviewer #1: Yes

3. Have the authors made all data underlying the findings in their manuscript fully available?

Reviewer #1: Yes

4. Is the manuscript presented in an intelligible fashion and written in standard English?

Reviewer #1: Yes

**Reviewer #1:**  I have no major comments on this manuscript. The claims are adequately supported by the data, the discussion is reasonably nuanced, and better knowledge of Symsagittifera's sensory preferences is a useful addition to the literature.

Minor comments:

- Not sure how feasible it is to do additional experiments for this project, so if that isn't possible, it might be worth discussing the negative results in more detail. Specifically: why did you choose 20mM for amino acids? How might the results be different at different concentrations? Might different AAs diffuse through the agarose at different rates?

- Along similar lines, perhaps the concentrations of the symbionts were too high, causing an aversive response that might be context-dependent?

- You speculate that immature animals may find the symbionts attractive. Is it possible to do this experiment?

- Perhaps it would help to add some context to the discussion - eg by explicitly comparing your results to expectations or relevant data from previous work, especially Thomas et al. 2023

**Do you want your identity to be public for this peer review?** For information about this choice, including consent withdrawal, please see our Privacy Policy

Reviewer #1: No

---

## [Author Response · Author response to Decision Letter 1]

26 May 2025

Journal Requirements:

All formatting has been amended to comply with guidelines

We have added a permits and site access section to the methods which reads:

Permits and site access

No permits were required for this work on invertebrate model animals. Site access to the Roscoff Marine Biology Station was provided and supervised by Xavier Bailly (xavier.bailly@sb-roscoff.fr)

grants:

-310030_219348

-IZKSZ3_218514

to SGS by the Swiss National Science Foundation

We would like to express our gratitude to the Swiss National Science Foundation (SNSF) for providing the funding for this project (grant 310030_219348 and IZKSZ3_218514 to SGS). We are also thankful to the University of Fribourg (UniFr), members of the Department of Biology and the Marine Station in Roscoff for their support and help throughout our research. We also thank all members of the Sprecher Lab for their invaluable feedback on our experimental design, procedures, and reporting.

grants:

-310030_219348

-IZKSZ3_218514

to SGS by the Swiss National Science Foundation

We have uploaded the data to the Zenodo repository:

https://zenodo.org/records/15470219?token=eyJhbGciOiJIUzUxMiJ9.eyJpZCI6ImRiYTc3OTM0LWNkMDItNDVkMC04NzY4LWE0N2FiNDQ0NjNhOSIsImRhdGEiOnt9LCJyYW5kb20iOiJiYWQ5ZmU2NGI3ODdhMzM5YzhhY2I1ODUzODE1YmFhZSJ9.KJhEgc7RN0yJZeKa9npMKW3wBQx8lxdL_XoUNSqaPo8gY6xI2Djula4wxGVZrwfyDPuLm9oTamO9RGZXOahk4w

Reviewers' comments:

Reviewer's Responses to Questions

Comments to the Author

1. Is the manuscript technically sound, and do the data support the conclusions?

Reviewer #1: Yes

2. Has the statistical analysis been performed appropriately and rigorously?

Reviewer #1: Yes

3. Have the authors made all data underlying the findings in their manuscript fully available?

Reviewer #1: Yes

4. Is the manuscript presented in an intelligible fashion and written in standard English?

Reviewer #1: Yes

5. Review Comments to the Author

Reviewer #1: I have no major comments on this manuscript. The claims are adequately supported by the data, the discussion is reasonably nuanced, and better knowledge of Symsagittifera's sensory preferences is a useful addition to the literature.

We thank the reviewer for these kind words

Minor comments:

- Not sure how feasible it is to do additional experiments for this project, so if that isn't possible, it might be worth discussing the negative results in more detail. Specifically: why did you choose 20mM for amino acids? How might the results be different at different concentrations? Might different AAs diffuse through the agarose at different rates?

Thank you for this comment. Indeed, at the moment it would not be feasible to perform additional experiments, however we believe that in the context of this study the questions are adequately investigated.

Regarding amino acid concetnration, we have added the relevant text in the discussion section which reads:

The 20mM concentration for amino acids was chosen as it is the standard for small animal behaviour, such as in the Drosophila larva, though it has been used in vertebrates as an average concentration where they are responsible (for instance in spider monkeys) References are added to the text. It has been shown to provide sufficient responses without being toxic. While higher than the physiological concentrations or ones found in regular food, the diffusion through agarose in general is slower than in water. We do not believe that diffusion rates will differ between different amino acids, as the average size is around 4Å, while agarose gel pores typically range from between 30-200nM. Other factors, such as ion concentrations may be more relevant, thus for our experiments concentrations were kept equal.

- Along similar lines, perhaps the concentrations of the symbionts were too high, causing an aversive response that might be context-dependent?

These concentrations of symbionts are the same that we are using in the laboratory (in fact, in the laboratory of Xavier Bailly) to maintain stable cultures of S. roscoffensis, able to close their life cycle. They follow similar concentrations used by Provasoli and Douglas in the past (see next comment). In this context we would call those concentrations “physiological”, in the sense that they work well and don’t produce any disruptive response.

- You speculate that immature animals may find the symbionts attractive. Is it possible to do this experiment?

- Perhaps it would help to add some context to the discussion - eg by explicitly comparing your results to expectations or relevant data from previous work, especially Thomas et al. 2023

-Correct. This experiment has been done in different laboratories, where the algae has been added to the aposymbiotic animals (hatchlings). Provasoli et al. (1968), Douglas (1983) and Arboleda et al (2018) have done, and reported, the positive results of this experiments. One of us, Dr. Xavier Bailly, has developed stable cultures of S. roscoffensis and demonstrated (unpublished, though) that these algae+host mixtures in culture are perfectly viable and hasn’t observed any toxic effect. In fact, the possibility of mixing algae and aposymbiotic hosts allowed him to maintain cultures for three years in a row (without any problem).

-Our work represents a good complement to the one published by Thomas and collaborators, where they test the response of S. roscoffensis to several physical and chemical cues. Though they use the same species, their stimuli are different than those studied by us (light, vibration, sea water or frozen worms). This complementary approach has now been stressed in the text. Perhaps the most relevant issue here, which merits further experimentation, is the response to a frozen worm versus a worm-extract. Our negative response might be the result of chemicals extracted from the animals during the maceration process. It is possible that their repellent nature would be a natural response of worms in a situation where the environment signals stress or mortality to the living specimens, thus generating a flight response from the “mortality cues”.

6. PLOS authors have the option to publish the peer review history of their article (what does this mean?). If published, this will include your full peer review and any attached files.

Do you want your identity to be public for this peer review? For information about this choice, including consent withdrawal, please see our Privacy Policy.

Reviewer #1: No

---

## [Decision Letter · Decision Letter 1]

4 Jul 2025

Dear Dr. Sprecher,

Thank you for submitting your manuscript to PLOS ONE. After careful consideration, we feel that it has merit but does not fully meet PLOS ONE’s publication criteria as it currently stands. Therefore, we invite you to submit a revised version of the manuscript that addresses the points raised during the review process.

We look forward to receiving your revised manuscript.

Kind regards,

Phuping Sucharitakul

Academic Editor

PLOS ONE

Journal Requirements:

Reviewers' comments:

Reviewer's Responses to Questions

**Comments to the Author**

Reviewer #2: All comments have been addressed

Reviewer #3: (No Response)

2. Is the manuscript technically sound, and do the data support the conclusions?

Reviewer #2: Yes

Reviewer #3: Partly

3. Has the statistical analysis been performed appropriately and rigorously?

Reviewer #2: Yes

Reviewer #3: I Don't Know

4. Have the authors made all data underlying the findings in their manuscript fully available?

Reviewer #2: Yes

Reviewer #3: No

5. Is the manuscript presented in an intelligible fashion and written in standard English?

Reviewer #2: Yes

Reviewer #3: No

Reviewer #2: I have not reviewed the first version of this manuscript but I was asked to gauge in on the revised version and I think the authors have addressed all the reviewers' suggestion and produced a sound manuscript.

Reviewer #3: (No Response)

**Do you want your identity to be public for this peer review?** For information about this choice, including consent withdrawal, please see our Privacy Policy

Reviewer #2: **Yes: ** Giorgio Gilestro

Reviewer #3: No

---

## [Author Response · Author response to Decision Letter 2]

4 Aug 2025

We acknowledge the referees for their invaluable input towards improving this manuscript. Their suggestions have undoubtedly elevated the original manuscript to be more thorough, accurate, and informative. Thank you.

General comments

In this paper, the authors investigated the chemosensory (salinity, amino acids and algae) and thermosensory (temperature) capacity of wild-caught acoel Symsagittifera roscoffensis in a laboratory setting. This is an exploratory study to determine whether these worms are able to detect and respond to these different stimuli. The results show that Symsagittifera roscoffensis is able to detect both chemical and physical cues, but at different levels. Only the amino acid Trytophan initiated a significant response. Only the 68 ppm salt concentration initiated a significant response. According to a following test, the initiated response seems to be aversion. For algae, F/2 medium seems to have elicited a preference response, while T. convolutae, C. concordia and T. chuii an avoidance response. According to a following test, the initiated response for the symbiont T. convolutae seems to be aversion. Finally, only the presence of high temperature elicited a response from the worms. In all cases, the worms seemed to avoid high temperatures.

Chemical and sensory ecology are important and growing fields, and I'm delighted to see the authors' interest in these still little-explored research fields. This exploratory study will help expand knowledge in these two fields, which is particularly important in the context of climate change, where aquatic environments are particularly affected. The study system chosen is very interesting and original. I applaud the authors' boldness in focusing on a lesser-known group of animals that are nonetheless the foundation of trophic webs.

However, I have a few issues that need to be addressed before publication. I also offer suggestions for improving the manuscript.

*Next time, I strongly recommend to include line numbers to facilitate communication.

Thank you for this suggestion, we have included line numbers in this revised version. The line numbers present in this response are in relation to the “track changes” version of the manuscript.

Major issues

The experiments carried out seem satisfying. However, the manuscript lacks scientific rigor. Several corrections need to be applied to meet publishable scientific quality standards.

1. The problematics/research question, aims, hypotheses, method and results should all be clearly aligned. At present, it is difficult to understand the study before reading the results. I recommend restructuring the ideas and making sure that the method (protocol) and results address the research question and objectives. A clear statement of whether the objectives have been met and whether the hypotheses are supported by the results should also be present. Without restricting authors to the classic writing format, the minimum requirements for assessing the validity of a study must be met.

Thanks for the suggestion. Yes, we are aware that the introductory paragraph was, most probably, unclear and lacking a proper explanation of the study context. We have added the following paragraph for clarity:

“The phylogenetic position of acoels, along with the fact that some species live in close association with algae, has made them of particular interest in the fields of evolutionary developmental biology (evo-devo) and comparative physiology. However, most existing studies on these animals focus on developmental and anatomical aspects, with relatively few addressing their physiology or the influence of environmental conditions on behavior, survival, or the dynamics of the host–algae association. To address this gap, we initiated a series of experiments aimed at testing the effects of various chemicals on these animals. Most of the compounds we tested—such as variations in salinity and temperature—are highly relevant to the rapidly changing conditions in coastal environments, where many acoel species naturally occur.”

1. The authors mention that ASW (34ppm) is aversive. However, this is not what is measured in the experiment. It's possible that the worms simply prefer the non-salty side without necessarily avoiding the other. I suggest that the authors add some nuance to their statements and don't mix up the results with interpretation. The PI (result) indicates that more worms chose the non-salty side. The interpretation is that the salty side is aversive

We thank the reviewer for pointing out this lack of clarity. We have thus adjusted the results section involving these experiments to read as follows (Lines 166-167): “The finding that S. Roscoffensis appears to prefer lower salt concentrations suggests that this condition is preferential for the acoel”

We would like to point out, however, that the ‘aversion’ was further determined by the subsequent barrier assay experiments, where high salinity was able to negate the positive phototaxis behaviour

1 -salty side is prefered). Preference/avoidance is an interpretation of the results. This interpretation should therefore be accompanied by a "We suggest/propose that...". This is repeated for the presentation of each result. Since the results seem to be combined with the discussion, authors should check each result presented to ensure that the difference between the result and the interpretation is unambiguous.

We thank the reviewer for pointing this out. Indeed, the interpretation may appear somewhat liberal. We have adjusted the text in the following places to remedy this:

Line 133: ...largely match those of the plain agarose control, suggesting animals did not prefer either side

Line 138: we observed that there was a moderate preference across the longer timescale

Line 154: We observed one-fold ASW (34ppm) showed a mildly positive, albeit insignificant, preference index

Line 158: we observed that the PI for two-fold ASW concentration (68ppm) was significantly negative across the time series

Line 159: However, they show a slight but statistically insignificant avoidance negative PI of to T. striata extracts

Line 213: No difference in the side preference index

Line 215: no significant preference index

Line 218-219: S. roscoffensis the animals tend to prefer show a positive PI to the low temperature side

1. The discussion lacks references. Some paragraphs have 0 to 1 references. With less than 2 or 3 sources, it's difficult to have a deep analysis of the situation. It would be useful to use a variety of references and examples to deepen the ideas. If there are no references on this species, which seems understudied according to the authors, the examples may come from similar species or those living in similar environments.

We agree with the reviewer that the number of references is sparse, however this is, indeed, due to a severe lack of modern (<50 years) research conducted on this topic, in this scenario, in marine animals. We have now included, to our knowledge, all relevant references that are available, and in turn focused the discussion on exploring the potential significance of our results

The findings should also be put into a broad context. Does it support existing literature or not? If not, why not?

1. The discussion should contain the limits/caveat of the experiment. Scientific rigor requires the assessment of limits. The authors should describe some shortcomings that may affect data interpretation, e.g. :

What are the confounding factors? What are the differences between the laboratory and the natural environment? What could have been improved? What should be tested to verify interpretations? What lacked precision?

Thank you for pointing this out. We have adjusted the discussion section in order to address the reviewer’s concerns:

Line 347: Under laboratory conditions, i.e. agarose substrate vs natural sand, our findings indicate...

Lines 367-378: To isolate responses exclusively to the stimuli of interest, our experimental conditions therefore differ from the natural environment in which S. roscoffensis is typically found. For instance, the substrate used in our experiment was agarose, as opposed to the natural substrate of sand, which may contain a large variety of chemical and biological contaminants that could, in turn, affect the behaviour of the animals. However, for the purposes of this study, we aimed to observe any response specifically to a narrow range of individual, ecologically relevant stimuli, rather than observing the animals’ behaviour in the natural environment. Furthermore, we maintained a constant ambient temperature, which may not be the case in nature due to factors such as winds, waves, or clouds. Further studies may be performed to understand the impact of these factors on the responses of S. roscoffensis to varied stimuli. For example, future work could use sand as a substrate, perform chemosensory experiments under varying temperature and light conditions, or test at different points of the tidal cycle.

Without invalidating their study, the authors must discuss certain points, as no study is perfect.

The method is missing a section on statistical analysis. Without this section, it is not possible to assess the validity and reliability of the results and conclusions drawn by the authors. I will, however, make a few comments based on what is mentioned in the figure titles. Was Dunnet's T3 multiple comparisons test carried out for each time (2, 5, 10 min) separately?

Why did the authors choose these tests? Were the application conditions respected? These details should be included in the method.

Did a single worm do several trials (tyrosine, glutamine, etc.)? If so, were the worms always in the same group? If the groups remain the same, it should be added to the model as a random factor since the analyses are based on repeated measures (One-way repeated measures ANOVA or GLMM).

We thank the reviewer for pointing out this critical omission. We have now added a statistical analysis section to the methods:

Statistical analysis:

All statistical analysis was performed using GraphPad Prism 10. For each trial of each experiment, a fresh set of acoels was used, thus no worms were subjected to multiple stimulation experiments. For time trials, each time point was treated independently. Welch’s test was used to determine unequal variance within the individual groups, and, combined with a relatively low sample size (n=10), Dunnet’s T3 multiple comparison test was chosen. Individual statistical results are presented in the relevant results sections and figure legends.

Minor issues

1. I suggest that authors proofread the entire text to correct spelling, grammatical and wording errors. To facilitate correction, authors can read the text starting from the end, or ask a third party who was not involved in drafting to read the text. In addition, certain scientific words relating to animal behaviour are not used correctly. Authors need to revise certain definitions and ensure that the scientific terms used support exactly what they mean.

We thank the reviewer for pointing this out. We have ensured to correct a number of spelling and grammatical errors, including those found in scientific terms

1. Authors will have to modify the formatting of references to meet the journal's requirements :

References are listed at the end of the manuscript and numbered in the order that they appear in the text. In the text, cite the reference number in square brackets (e.g., “We used the techniques developed by our colleagues [19] to analyze the data”). PLOS uses the numbered citation (citation-sequence) method and first six authors, et al.

https://journals.plos.org/plosone/s/submission-guidelines#loc-introduction

We have now ensured that the reference style meets the journal requirement

Abstract (page 2)

1. In the second sentence, the authors mention that "understanding how distinct sensory cues elicit or modulate certain behaviour thus provides insights into the neuronal adaptations to rapid and continuous changes in the surrounding world". However, there is no reference to neuronal adaptation afterwards, although this seems to be an important subject of the study they way it is mentionned. I consider this paper to be in the field of animal behaviour rather than neuroscience. Authors should be careful about the scope of their claims.

We thank the reviewer for pointing this out, and have removed “neuronal” from the sentence in question to ensure that the scope of the study remains focused.

1. When stating the general problem (lack of knowledge on other sensory systems), I recommend giving a short explanation on why bridging the gap matters, to give more context and justify the purpose of the study.

We thank the reviewer for this suggestion, and have now added a section to the introduction to address this (lines 92-94): Bridging this gap in knowledge may uncover how non-visual sensory modalities govern how S. roscoffensis locates and retains its algal symbiont and selects microhabitats in a highly variable intertidal environment. Establishing these capacities also provides comparative data for the early evolution of multimodal sensory integration in bilaterians.

1. While keeping this section concise with the main result, I suggest going into a little more detail. The method and main results should be incorporated to quickly understand the study when first reading the abstract. I suggest integrating the approach and the main results in a single sentence (e.g. Using XXX method, we found YYY).

Thank you for this suggestion. The end of the abstract now reads (Lines 35-37): Using two-choice and barrier assays, our findings support that S. roscoffensis shows avoidance behaviours to increased temperature and high salinity, preferring cooler environments with lower salinity.

1. What does the main result reveals? The aim of the study seems to be to explore the yet unknown sensory systems of S. roscoffensis. What do the results reveal about this? Does it suggest that S. roscoffensis can detect chemical and thermal cues?

I suggest adding a final sentence or two about the wider context of the results. For example, but without limitation, authors could mention:

-What the results demonstrate;

-What are the advances/implications in relation to previous works;

- How does the results change the understanding of this field (sensory ecology or neuronal adaptations) / how significant are the results?

Since these additions are likely to increase the size of the abstract, I suggest cutting into the study backgroud, which currently takes up a lot of space compared to the results (i.e. everything before "Currently little is known about...").

Thank you for this suggestion. We have added the following sentence at the end of the abstract:

We demonstrate that early-branching bilaterians possess the sensory capacity to identify specific chemical and environmental stimuli, adding to the knowledge that

1. "is" should be changed for "are" in the following sentence : "When the acoel Symsagittifera roscoffensis is exposed…"

We have made this correction, thank you.

1. The study topic is truly interesting and captivating. I think it's worth extending the introduction a little to take the time to properly introduce the concepts and justify the aim of the study.

Thank you for your suggestion, as per the previous comment, we have extended the introduction to include more background and justification for the study.

Introduction

1. Page 3 : The first paragraph of the introduction should include at least one or two references. An interesting way of inserting them is to add examples to support the facts.

Thank you for this suggestion. We have added a recent reference to the review by Oteiza and Baldwin (2021) to point out that despite the variability the phenotypes are typically conserved (lines 47-49):

While many animals utilise external sensory organs to determine the nature of their surroundings, the compositional, morphological and anatomical organization of sensory organs varies greatly in different animal clades, including different, often unrelated genes being responsible for the same functions, indicating

---

## [Editor Report · Decision Letter 2]

6 Aug 2025

Sensory discrimination of chemical and temperature stimuli in the acoel Symsagittifera roscoffensis

PONE-D-25-05435R2

Dear Dr. Sprecher,

I am pleased to inform you that your manuscript has been judged scientifically suitable for publication and will be formally accepted for publication once it meets all outstanding technical requirements.

Kind regards,

Phuping Sucharitakul

Academic Editor

PLOS ONE
---

## [Editor Report · Acceptance letter]

PONE-D-25-05435R2

PLOS ONE

Dear Dr. Sprecher,

I'm pleased to inform you that your manuscript has been deemed suitable for publication in PLOS ONE. Congratulations! Your manuscript is now being handed over to our production team.

Kind regards,

on behalf of

Dr. Phuping Sucharitakul

Academic Editor

PLOS ONE